# Knowledge, attitudes and acceptance of COVID-19 vaccine among pregnant women in Mbeya Region

Revocatus Lawrence Kabanga[1¤a]*, Vincent John Chambo[1¤b], Rebecca Mokeha[1,2]

1 Department of Obstetrics & Gynecology, Mbeya Zonal Referral Hospital, Mbeya, Tanzania,
2 Department of Obstetrics & Gynecology, University of Dar es Salaam-Mbeya College of Health and Allied Sciences, Mbeya, Tanzania

¤a Current address: Department of Internal Medicine, Mbeya Zonal Referral Hospital, Mbeya, Tanzania
¤b Department of Internal Medicine, Mwananyamala Regional Referral Hospital, Dar es Salaam, Tanzania
* revocatuskabanga2@gmail.com

## Abstract

COVID-19 has caused about 580 million cases and 6.4 million deaths worldwide by August 8th, 2022, including 8.7 million cases (173,063 deaths) in Africa. East Africa reported 1.39 million cases on July, 2022. Tanzania confirmed 37,865 cases and 841 deaths by 8th August 2022. Although billions of vaccine doses administered globally, just 17.6% of Tanzanians are fully vaccinated. Symptomatic pregnant women face a mortality risk that is 70% higher than in non-pregnant women.. Therefore, this study aimed at assessing knowledge, attitude, and acceptance of COVID-19 vaccine among pregnant women in the Mbeya region. A descriptive cross-sectional study was conducted in the Obstetrics and Gynecology department of MZRH. Three scores were calculated for participants' knowledge, attitude, and acceptance to COVID-19 vaccination. These scores were compared to many sample factors using binary logistic regression and the chi-square test. The study recruited 233 participants. Most participants (31.3%) relied on social media for Covid-19 vaccine information. Poor Covid-19 vaccine knowledge (71.2%), negative attitudes (76.8%), and low acceptance rate (38.6%) were observed. Multivariate analysis showed that greater acceptance was positively associated with having a chronic illness (AOR = 3.21, CI 1.448-7.123, P = 0.004), stronger vaccine attitudes (AOR = 1.26, CI 1.149-1.368, P = 0.015), better vaccine knowledge (AOR = 2.70, CI 2.587-2.810, P = 0.005), and prior vaccination history (AOR = 0.13, CI 0.068-0.183, P = 0.000). Conversely, preference for natural immunity (AOR = 0.42, CI 0.341-0.498, P = 0.018), and not yet being vaccinated (AOR = 0.67, CI 0.594-0.755, P = 0.000) were all linked to lower acceptance. Pregnant women exhibited low knowledge, attitude, and acceptance to COVID-19 vaccines. Misinformation about the COVID-19 vaccine causes pause. Education on COVID-19 vaccination is needed to enhance vaccine uptake among pregnant women. This group must comprehend COVID-19 immunization importance, safety, and efficacy.

**Data availability statement:** All relevant data are within the paper and its Supporting Information files.

**Funding:** The author(s) received no specific funding for this work.

**Competing interests:** The authors have declared that no competing interests exist.

## Introduction

### Background

Severe acute respiratory syndrome coronavirus 2 (SARS-CoV-2), the RNA virus that causes coronavirus disease 2019(COVID-19), also known as SARS-CoV-2 infection, was first diagnosed in Wuhan, China, in December 2019. Due to the rapidly escalating numbers of new infections outside China in less than three months, COVID-19 was declared a pandemic by the World Health Organization (WHO) on March 11, 2020, in a message delivered by Dr. Tedros Adhanom Ghebreyesus, the WHO Director-General [1]. As of August 8, 2022, the WHO recorded almost 580 million confirmed cases of COVID-19, resulting in 6.4 million fatalities across 213 countries [2].

The COVID-19 pandemic impacted numerous countries globally, including those in Africa. Africa was the last continent affected by the pandemic and was anticipated by researchers to be the region where the disease would spread rapidly and exert significant impact [3,4]. As of August 4, 2022, Africa has recorded over 8.7 million confirmed cases, resulting in 173,063 deaths across 47 countries [2]. As of July 2022, the cumulative number of recorded COVID-19 cases in East Africa was 1.39 million, with Ethiopia and Kenya being the most impacted nations [2].

The United Republic of Tanzania has also been impacted by the COVID-19 pandemic both economically and socially. The country has experienced three waves of the pandemic, with an increased impact of subsequent waves [5]. From 16th March 2020, when the initial case was reported, to 6:24 pm CEST on 8th August 2022, there have been 37,865 confirmed cases of COVID-19 and 841 documented deaths [2].

The COVID-19 pandemic has caused widespread disease and death globally, with the absence of a vaccination significantly contributing to elevated morbidity and mortality rates. COVID-19 vaccines are now being distributed and made accessible in many different countries [6].

As of 01 August 2022, over 12.4 billion vaccine doses have been provided globally, with more over 4.88 billion individuals fully immunized [2]. As of August 1, 2022, Africa had received over 924 million vaccination doses and delivered more than 639 million doses [7].

Initially, The United Republic of Tanzania exhibited anti-vaccination buzz [8] however adopted the vaccination strategy [9], with the first consignment of 1,058,450 doses of Johnson & Johnson COVID-19 vaccines in July 2021, and 1,065,600 doses of Sinopharm vaccines supplied by the Chinese government through the COVID -19 Vaccines Global Access (COVAX) facility in early October 2021 [10]. As of July 24, 2022, more than 16.8 million doses had been administered, with over 10.5 million individuals fully vaccinated [2].

Pregnant women constitute a distinct demographic with an increased susceptibility to COVID-19 morbidity and mortality, possibly experiencing a more severe progression of the disease compared to their non-pregnant counterparts [11]. Symptomatic pregnant women face a 70% elevated risk of mortality compared to their non-pregnant symptomatic peers [12]. Numerous arguments exist regarding the safety and efficiency of COVID-19 immunizations in pregnant women; nonetheless, the COVID-19 vaccination in this group is as successful as in the general population, providing dual advantages to both mothers and newborns [10].

The United Republic of Tanzania employed a multifaceted approach to promote COVID-19 vaccination including "Mziki Mnene" campaign translating to "Heavy Music," was a dynamic COVID-19 vaccination outreach initiative that effectively combined music, community engagement, and public health services to accelerate vaccine uptake [13]; Organized by the EpiC project in collaboration with E-FM radio and supported by USAID attracted large crowds and provided on site vaccination. The campaign's slogan, "Bega kwa bega, ujanja kujanja" translating to "Shoulder to shoulder, vaccination is intelligence," aimed to reshape public perception and encourage vaccine acceptance [14]. Others included SMS outreach, multimedia engagement, mobile vaccination units and door to door vaccination services [15].

Despite various initiatives to combat the disease via vaccination, reports indicated that only 17.6% of Tanzanians were fully vaccinated, presenting a significant challenge to the global effort to control the COVID-19 pandemic, as the virus is rapidly mutating in association with successive waves of outbreaks [2].

Tanzanians, similarly to several other Africans, were perceived as at risk of under-immunization prior to the COVID-19 pandemic, exhibiting low levels of vaccination uptake and trust [16]. Furthermore pregnant women are known to be at significantly higher risk of severe COVID-19 related complications compared with non-pregnant women. Hence, protecting pregnant women against COVID-19 is critical [12].

Pregnant women are one of several vulnerable groups to COVID-19 vaccine disinformation particularly anti-vaccination droning as well as lack of reliable information due to social marginalization and language barriers [17]; Thus the purpose of this study was to assess knowledge on COVID-19 vaccination, attitudes to COVID-19 vaccines and acceptance towards COVID-19 vaccines among pregnant women in Mbeya region, Tanzania so as to create increased awareness related to the importance of COVID-19 vaccine as essential to expand vaccine utilization [18].

The World Health Organization declared COVID-19 as a pandemic on March 11, 2020, impacting over 200 countries [1]. This highly contagious disease has imposed a significant burden on the world, resulting in millions of cases and fatalities [2]. Additionally, the pandemic has triggered tremendous socio-economic and psychological impacts [19].

To curb the spread of the virus, billions of COVID-19 vaccines have already been administered worldwide [2]. However, an effective immunization campaign necessitates sufficient knowledge and a positive attitude toward the vaccine. A lack of knowledge and unfavorable attitude may result in delayed decisions regarding vaccination acceptance or outright refusal, despite the availability and accessibility of immunization services [20,21]. Disinformation regarding COVID-19 vaccines also appeared to play a major role in vaccination reluctance [22]. Enhancing public knowledge on COVID-19 vaccination is therefore a key strategy to boost the vaccine uptake [23].

Despite Tanzania ministry of health identification and WHO recommendation, the level of knowledge, attitude and acceptance of COVID-19 vaccines among pregnant women remain poorly understood. A study conducted in Tanzania indicated that approximately 65% of the people were reluctant to get vaccinated against COVID-19 [22]. However, this single study did not focus specifically on pregnant women nor did it determine the exact extent of vaccine acceptance within this vulnerable group. This gap highlights the need for targeted research to assess the knowledge, perceptions and acceptance of COVID-19 vaccines among pregnant women in Mbeya region.

### Objectives

**Broad objective.** To assess the current awareness on COVID-19 vaccine among pregnant women in Mbeya region.
**Specific objectives.**

- To determine the level of knowledge on COVID-19 vaccine among pregnant women in Mbeya region.

- To assess the attitude towards COVID-19 vaccines among pregnant women in Mbeya region.

- To determine COVID-19 vaccine acceptance and its associated factors among pregnant women in Mbeya region.

**Hypothesis.**

I. A lack of knowledge and negative attitudes lead to lower acceptance rate of COVID-19 vaccine among pregnant women in Mbeya region.

II. Disinformation regarding COVID-19 vaccine contributes to its lower acceptance rate among pregnant women in Mbeya region.

**Significance of the study.** The outcome of this study will determine the Knowledge, Attitude and Acceptance levels among pregnant women towards COVID-19 vaccine in Mbeya region as the road to create increased awareness of the importance of vaccination, responding to the hot spots of vaccine hesitancy, promoting vaccine utilization and thence establishing herd immunity as well as alleviating severe pandemic situation and improving maternal and newborn health.

## Methods

### Ethical issues

The permission to conduct the research was gained by approval letter from the University of Dar es Salaam Mbeya College of Health and Allied Sciences ethical committee, which was presented to the Mbeya area and Mbeya Zonal Referral Hospital administration. With the assistance of the medical professional in charge of antenatal care unit, we were able to reach the target population.

Each participant was educated about the research's purpose, process, benefits, and risks and they signed an informed consent form. Participants preserved the right to withdraw from the study at any point during the proceeding of the study. Furthermore, the anonymity of the information revealed was considered by using the information provided solely for research purposes, assigning numbers instead of actual participant names and data obtained were stored in a password-protected database (computer).

### Study design

An institutional based descriptive cross sectional research study was conducted from 01st September 2022–06th September 2022 at Mbeya Zonal Referral Hospital (MZRH).

### Study area

This study was carried out in Mbeya Zonal Referral Hospital, Obstetrics and Gynecology Section, located in Mabatini Ward, Mbeya Urban District, Mbeya Region, Tanzania. It is located at 08°54'21'' south and 33°25'59'' east, surrounded by Itigi ward on the south, Nzovwe ward on the east, Sisimba ward on the north, and Mbalizi road ward on the west. The hospital serves about 8 million people throughout six southern highlands regions: Katavi, Njombe, Rukwa, Ruvuma, Iringa, and Mbeya. It not only provides advanced medical services but also engages in research and training, attracting health care professionals dedicated at improving public health outcomes. Mbeya serves as a strategic point for cross-border collaborations and initiatives, especially in public health and trade.

### Study population

The study population comprised of pregnant women who attended antenatal clinic (ANC) at Mbeya Zonal Referral Hospital during the study period.

### Eligibility

**Inclusion criteria.** Pregnant women aged 18 years and above who attended antenatal clinic at MZRH during the study period were included.

**Exclusion criteria.** Pregnant women aged 18 years and above who were seriously ill, those with communication barriers (unable to hear/read Swahili or English and those diagnosed to have mental illness (documented by a doctor) were excluded from the study.

**Sample size estimation and sampling procedure.** In this study, the sample size required for this study was determined using a formula of cross sectional study by Kish Leslie [24]

$$n = \frac{Z^2 p(1-p)}{e^2}$$

Where

$n$ = was the sample size estimated, p = 18.6%, proportion for COVID-19 vaccine acceptance among pregnant women from a study in Ethiopia [25], Z = 1.96 which corresponds with 95% confidence interval, e = 0.05 marginal error

Then;

$$n = \frac{1.96^2 0.186(1-0.186)}{0.05^2}$$

$n$ = 233

Therefore, the sample size required in this study was 233 participants.

A simple random sampling technique was used to select pregnant women. Sampling frame was developed from ANC register from the facility. The total population of the study from the facility was about 600 pregnant women, who were used to calculate the sampling interval required to enable systematic sampling during data collection for those who met inclusion criteria

The study population from the facility was divided by 233 sample size to get a sampling interval

i.e., ($i = N/n = 600/233 = 2.6 \approx 3$)

Hence, the sampling interval was every 3rd pregnant woman

Where

$N$ = 600 (Total population for pregnant women attending ANC at Mbeya Zonal Referral Hospital)

$n$ = 233(Sample size for the study)

$i$ = 3 (interval during data collection)

**Data collection tools and procedure.** Primary data was collected using a structured questionnaire from the target population sampled (pregnant women) (S1 Data).

The hospital director sought an approval letter for data collection in the Mbeya Zonal Referral Hospital. Permission was sought from the unit in charge at the ANC clinic, and the pregnant women were approached. The researcher explained to them the purpose of the study and to get their informed consent. Those who consented were asked to fill out the questionnaires. A systematic sampling technique was employed to prevent bias during data collection. To achieve this the facility had at least 80 pregnant women who attended ANC clinic daily. The researcher took 1 week to collect data due to financial crisis required to extend the process more. Therefore, each day the researcher could get a minimum of 33 respondents to participate in the study. The selection was made for every 3rd pregnant woman who arrived from home for the ANC clinic that day and met the study's inclusion criteria. Likewise, was asked for her consent to participate in the study with the assistance of ANC nurse. Then data were collected from pregnant women in the waiting queue to enter the doctor's consultation room.

**Questionnaire.** The questionnaire consisted of four major sections: socio-demographic data and obstetric features, awareness of COVID-19 vaccination, attitude toward COVID-19 vaccine, and acceptance of COVID-19 vaccine.

**Validity and reliability.** To ensure the study's validity, the questionnaire was prepared based on the literature review and study objectives reflecting standard survey for vaccine hesitancy, assessed by the supervisor, and pre-tested before being introduced into the field for data collection.

The researcher tested the reliability by pre-testing the questions on pregnant women attending at Mbeya Zonal Referral Hospital to ensure respondents' straightforward questions were understood. The questions could respond and measure what the study wanted to achieve.

Additionally, the reliability of the knowledge, attitude, and acceptance questionnaires was evaluated, with Cronbach's alpha values of 0.81, 0.82, and 0.92, showing acceptable internal consistency (S1 Text).

The questionnaire was structured and translated into Swahili to prevent misinterpretation of the questions during the interview for optimal reliability. Translation to Swahili language was necessary as most of the population are aware of the language to avoid language barriers and misunderstanding thus optimizing the outcomes

**Data processing and analysis.** The data was validated for completeness, coded, cleaned, and entered into Epi-Data version 7 before being exported to the Statistical Package for Social Science (SPSS) version 27 for analysis. Data were summarized using mean and standard deviation for numerical variables and frequency and percentage for categorical variables. Data were presented in the form of tables, figures, and narratives. The factors related with COVID-19 vaccine uptake among pregnant women in the Mbeya region were investigated using binary logistic regression and the Chi-square test (S2 Text).

## Definition of variables

### Dependent variables

The basic outcomes of this study (dependent variables) were: -

**Having good knowledge of COVID-19 vaccine, moderate or poor.** To assess the participants' knowledge about the COVID-19 vaccination, nine related questions were posed to them. Each correct answer received a score of one, while incorrect responses received a score of zero. Using Bloom's cut off point, respondents' overall knowledge was classed as good if their score was between 80 and 100%, moderate if their score was between 60 and 79%,and low if their score was less than 60% [26].

**Having positive attitude of COVID-19 vaccine, neutral or negative.** Nine related questions were used to assess the participants' attitudes toward the COVID-19 vaccination. A positive response received a one-point score, and a negative response received zero points. Bloom's cut off point was used to categorize attitudes towards the COVID-19 vaccine as positive (80–100%), neutral (60–79%), or negative (< 60%) [27].

**Acceptance of COVID-19 vaccine or not.** Four related questions were asked to the participants, a specific item "If COVID-19 vaccine were recommended for pregnant women, would you get vaccinated?" those who responded "YES" for this question were considered as vaccine acceptance and those who responded "NO" for this question were regarded as vaccine hesitancy [28].

### Independent variables

These included maternal age, level of education, occupation, marital status, religion, gravidity, parity, gestational age, chronic medical illness(those with specific conditions like heart disease, chronic kidney disease, epilepsy, sickle cell disease, diabetes mellitus, chronic hypertension etc.), high risk pregnancy(one in which the health of mother, fetus or both is at greater risk than usual like age > 35 years, pre-existing medical conditions, multiple pregnancy, gestational diabetes, pre-eclampsia, bad obstetric history etc.), number of people in the household, number of school age children in the household.

## Results

### Background characteristics of the participants

The sociodemographic characteristics are summarized in (Table 1). A total of 233 pregnant women were interviewed giving a response rate of 100%. The median age of respondents was 30 years (IQR: 26, 34). One hundred forty-eight

**Table 1. Showing Sociodemographic characteristics of the participants.**

| Variable | Frequency | Percentage (%) |
|---|---|---|
| **District name** | | |
| Mbeya urban | 148 | 63.52 |
| Chunya | 43 | 18.46 |
| Mbarali | 22 | 9.44 |
| Kyela | 14 | 6.01 |
| Rungwe | 6 | 2.58 |
| **Maternal age (in years)** | | |
| 18−23 | 28 | 12.02 |
| 24−29 | 81 | 34.76 |
| 30−35 | 85 | 36.48 |
| 36−41 | 32 | 13.73 |
| 42−47 | 7 | 3.00 |
| **Marital status** | | |
| Married | 147 | 63.09 |
| Others* | 86 | 36.91 |
| **Education level** | | |
| No formal education | 22 | 9.44 |
| Primary education | 30 | 12.88 |
| Secondary education | 110 | 47.21 |
| College and above | 71 | 30.47 |
| **Occupation** | | |
| Employed (HCW, teacher, lawyer) | 66 | 28.33 |
| Farmers | 57 | 24.46 |
| Self-employed (Petty traders) | 44 | 18.88 |
| Unemployed | 66 | 28.33 |
| **Religion** | | |
| Christian | 124 | 53.22 |
| Muslim | 105 | 45.06 |
| None | 4 | 1.72 |
| **Number of people in household** | | |
| <5 | 139 | 59.66 |
| ≥5 | 94 | 40.34 |
| **Number of children in household** | | |
| <5 | 226 | 97.00 |
| ≥5 | 7 | 3.00 |

*(Single, divorced and widowed), HCW Health care workers

(63.52%) were from Mbeya urban district, most of the participants (63.09%) were married, while 110(47.21%) had secondary school education. Compared to other occupations, the majority of participants 66(28.33%) were equally employed and unemployed.

Health status and obstetric characteristics: One hundred forty (60.09%) of participants were in the third trimester, whereas 203(87.12%) were free of any chronic medical problem. The majority of respondents, 174(74.68%), did not have a high-risk pregnancy. Approximately 67 (28.76%) of the study participants were nulliparous (Table 2).

**Table 2. Showing obstetric characteristics of the participants.**

| Variable | Frequency | Percentage (%) |
|---|---|---|
| **Gestational age (in weeks)** | | |
| ≤13 | 7 | 3.01 |
| 14 − 27 | 86 | 36.91 |
| ≥28 | 140 | 60.09 |
| **Gravidity** | | |
| <5 | 212 | 90.99 |
| ≥5 | 21 | 9.01 |
| **Parity** | | |
| Nulliparous | 67 | 28.76 |
| <5 | 163 | 69.96 |
| ≥5 | 3 | 1.28 |
| **Chronic medical illness** | | |
| No | 203 | 87.12 |
| Yes | 30 | 12.88 |
| **High risk pregnancy** | | |
| No | 174 | 74.68 |
| Yes | 59 | 25.32 |

## Source of information on COVID-19 vaccine

The majority of participants (76%) had heard about the COVID-19 vaccine (Table 3). Social media was the most commonly reported source of information on the COVID-19 vaccine, accounting for 31.33% (Fig 1).

## Knowledge of COVID-19 vaccination

Table 3 presents the knowledge results for the COVID-19 vaccination. The majority of participants (71.24%) had little understanding of the COVID-19 vaccination. When asked if there is a cure for COVID-19, 147 (63.09%) participants responded "Yes," whereas most participants (75.97%) had heard about the COVID-19 vaccine, and the majority of respondents (76.39%) were aware that COVID-19 immunization had begun in Tanzania.

More than half of participants (51.50%) did not have enough knowledge regarding the safety of the COVID-19 vaccine, while 51.93% did not have enough evidence on the safety of the COVID-19 vaccine for pregnant women. One hundred forty-three (61.37%) respondents indicated that the COVID-19 vaccination may cause COVID-19, while 163 (69.96%) claimed that the newly discovered COVID-19 vaccine differed from existing vaccines. When asked if the COVID-19 vaccine can induce infertility, more than half of the respondents (56.22%) said "yes," although the majority of participants (77.68%) emphasized the necessity of other COVID-19 prevention methods. Thus, generally, interviewed individuals had "poor" understanding of the COVID-19 vaccination (Fig 2).

## Attitude towards COVID-19 vaccine

One hundred and seventy-nine (76.82%) of those interviewed had a negative attitude regarding COVID-19 vaccinations. More over half of the participants (60.94%) believed that the government of the United Republic of Tanzania made decisions in their best interests regarding COVID-19 vaccine. Approximately 37.77% of respondents agreed that the COVID-19 vaccine was necessary for pregnant women, while 66.95% and 58.80% were concerned that the vaccine might harm their bodies and unborn babies, respectively. (Table 4)

**Table 3. Knowledge of COVID-19 vaccine among participants.**

| Variable | Frequency | Percentage (%) |
|---|---|---|
| Currently, is there an effective cure of COVID-19? | | |
| Yes | 147 | 63.09 |
| No | 86 | 36.91 |
| Have you heard about COVID-19 vaccine? | | |
| Yes | 177 | 75.97 |
| No | 56 | 24.03 |
| Is COVID-19 vaccination started in Tanzania? | | |
| Yes | 178 | 76.39 |
| No | 55 | 23.61 |
| Does newly discovered COVID-19 vaccine differ from other vaccines? | | |
| Yes | 163 | 69.96 |
| No | 70 | 30.04 |
| Do you have enough information about COVID-19 vaccine and their safety? | | |
| Yes | 113 | 48.50 |
| No | 120 | 51.50 |
| Can COVID-19 vaccine cause COVID-19? | | |
| Yes | 143 | 61.37 |
| No | 90 | 38.63 |
| Do you have enough evidence about COVID-19 vaccine safety during pregnancy? | | |
| Yes | 112 | 48.07 |
| No | 121 | 51.93 |
| Do you think COVID-19 vaccine can cause infertility? | | |
| Yes | 131 | 56.22 |
| No | 102 | 43.78 |
| Even if there is a vaccine are other preventive measures very important? | | |
| Yes | 181 | 77.68 |
| No | 52 | 22.32 |

The majority of participants (63.95%) preferred natural immunity over the COVID-19 vaccine; only 27.47% of pregnant women were immunized against COVID-19; however, when asked to reconsider the decision to take the COVID-19 vaccine, more than half of participants (59.23%) agreed to reconsider COVID-19 vaccination. The overall opinion towards the COVID-19 vaccination was judged "negative."" (Fig 3).

## Acceptance to COVID-19 vaccine

A total of 93 respondents (39.91%) had received other vaccines within the past five years. At the time of survey, only 27.47% of pregnant women had already received the COVID-19 vaccine. When asked if they would accept the vaccine if it were recommended for pregnant women, 38.63% indicated willingness to be vaccinated. Similarly, 34.33% of participants expressed willingness to have their babies vaccinated against COVID-19 after delivery 9 (Table 5).

## Factors associated with acceptance of COVID-19 vaccine

In binary logistic regression, education level, religion, number of people in the household, chronic medical illness, high risk pregnancy, reliable source of information, trusting pharmaceutical companies, preference for natural

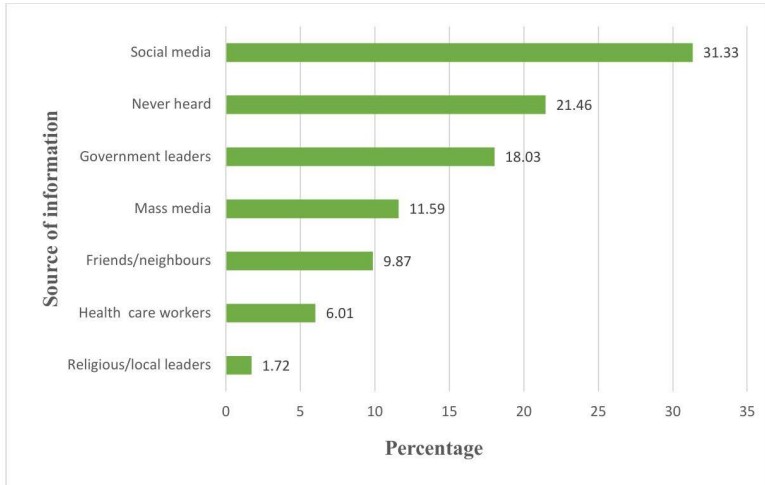

**Fig 1. Showing source of COVID-19 vaccine information.** Bars showing the percentage of pregnant women and the source the relying on for COVID-19 vaccine information.

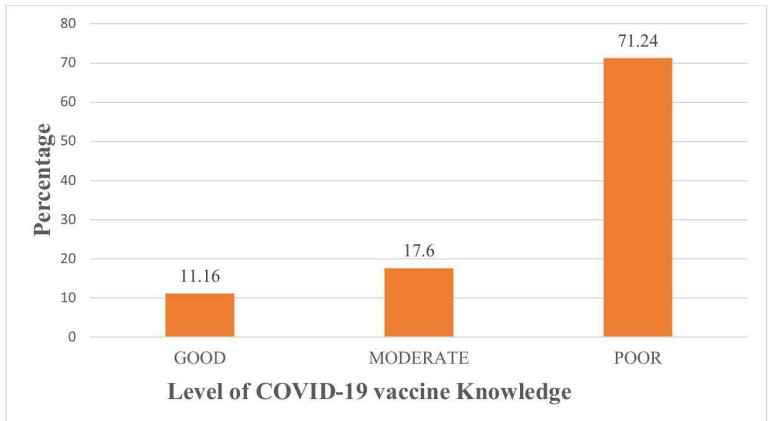

**Fig 2. Showing the level of knowledge on COVID-19 vaccine.** Bars showing the percentage of pregnant women with their level of COVID-19 vaccine Knowledge.

immunity over COVID-19 vaccine, having vaccinated for the past five years, attitude towards COVID-19 vaccine, and knowledge on COVID-19 vaccine were determinants of COVID-19 vaccine acceptability (P-values <0.05) (Table 6) (Table 7).

Higher education level translated to increased vaccine acceptance compared to those with primary and non-formal education (OR=0.35, CI 0.187-0.644, P = 0.001). Religion appeared to propagate vaccine skepticism leading to lower uptake (OR=0.53, CI 0.317-0.887, P = 0.016). Number of people in the house hold had equal likelihood of vaccine hesitancy (OR=0.49, CI 0.278-0.849, P = 0.011). Pregnant women with chronic medical illness had 3.21 folds of vaccine acceptance compared to those without chronic illness (AOR = 3.21, CI 1.448-7.123, P = 0.004).

Despite unique consideration, high risk pregnancy showed increased vaccine hesitancy compared to those who were not (OR=0.84, CI 0.455-1.553, P = 0.020). Information sought from social media had approximately thrice

**Table 4. Attitudes toward COVID-19 vaccine.**

| Variable | Frequency | Percentage (%) |
|---|---|---|
| Do you trust that your government is making decisions in your best interest concerning COVID-19 vaccination? | | |
| Yes | 142 | 60.94 |
| No | 91 | 39.06 |
| Do reports you hear/read make you reconsider the choice to take COVID-19 vaccine? | | |
| Yes | 138 | 59.23 |
| No | 95 | 40.77 |
| Do you trust pharmaceutical companies to provide credible data on COVID-19 vaccine safety & effectiveness? | | |
| Yes | 124 | 53.22 |
| No | 109 | 46.78 |
| Do the benefits of COVID-19 vaccine outweigh reported side effects? | | |
| Yes | 105 | 45.06 |
| No | 128 | 54.94 |
| Is COVID-19 vaccine essential for pregnant women? | | |
| Yes | 88 | 37.77 |
| No | 145 | 62.23 |
| Do you prefer natural immunity than COVID-19 vaccine? | | |
| Yes | 149 | 63.95 |
| No | 84 | 36.05 |
| Do you think COVID-19 vaccine can harm your body? | | |
| Yes | 156 | 66.95 |
| No | 77 | 33.05 |
| Do you think COVID-19 vaccine can harm your baby? | | |
| Yes | 137 | 58.80 |
| No | 96 | 41.20 |
| Is your decision to get vaccinated driven by your husband/partner? | | |
| Yes | 105 | 45.06 |
| No | 128 | 54.94 |

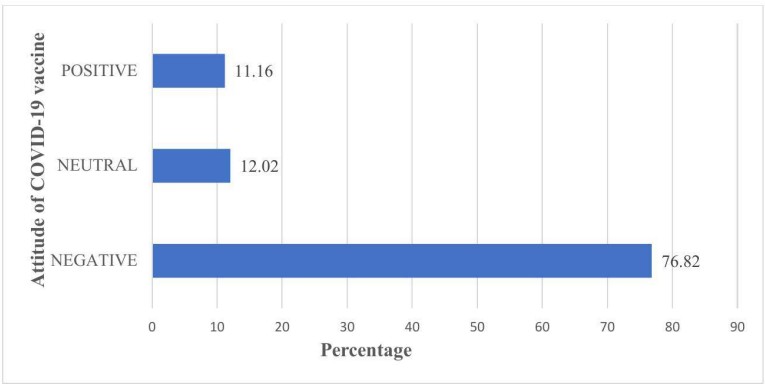

**Fig 3. Attitude level towards COVID-19 vaccine.** Bars showing the percentage of pregnant women and their attitude towards COVID-19 vaccine.

**Table 5. Acceptance to COVID-19 vaccine.**

| Variable | Frequency | Percentage |
|---|---|---|
| Have you been vaccinated for the last 5 years? | | |
| Yes | 93 | 39.91 |
| No | 140 | 60.09 |
| Have you been vaccinated against COVID-19? | | |
| Yes | 64 | 27.47 |
| No | 169 | 72.53 |
| If the COVID-19 vaccine is recommended for pregnant women, will you get vaccinated? | | |
| Yes | 90 | 38.63 |
| No | 143 | 61.37 |
| Are you going to have your baby vaccinated after birth? | | |
| Yes | 80 | 34.33 |
| No | 153 | 65.67 (%) |

likelihood of vaccine hesitancy compared to those sought from other sources (OR=2.53, CI 0.068-0.633, P = 0.006). Poor vaccine knowledge was linked to lower acceptance rate 2.7 folds compared to those with moderate and good knowledge (AOR = 2.70, CI 2.589-2.810, P = 0.005). Negative attitude towards COVID-19 vaccine had 1.26 folds of lower acceptance rate compared to those with neutral & positive attitudes (AOR = 1.26, CI 1.149-1.368, P = 0.015).

Skepticism towards pharmaceutical companies hindered vaccine acceptance to approximately 10 folds (OR=9.59, CI 4.992-18.431, P = 0.000). Additionally, preference to natural immunity contributed to vaccine hesitancy (AOR = 0.42, CI 0.341-0.498, P = 0.018), prior vaccination history in the last 5 years influenced higher acceptance rate (AOR = 0.13, CI 0.068-0.183, P = 0.000) and furthermore having being vaccinated against COVD-19 added more vaccine confidence and willingness when recommended compared those who were not vaccinated against Covid-19(AOR = 0.67, CI 0.594-0.755, P = 0.000).

A chi-square test was also used to investigate parameters linked with the observed Knowledge, Attitude and Acceptance for the COVID-19 vaccination. Fisher's exact test was used to identify significant variables, with a P-value <0.05 indicating statistical significance of the connections.

Poor knowledge of the COVID-19 vaccine was found to be connected with vaccine acceptance, with only 33.13% of pregnant women with low knowledge compared to 48.78% and 57.69% of those with moderate and good knowledge, respectively, having accepted the vaccine. (Chi = 7.88, P = 0.002) (Table 8).
Furthermore, only 6.02% of participants with poor knowledge of COVID-19 vaccine compared to 12.30% and 42.31% of those displayed moderate and good knowledge respectively had positive attitude towards COVID-19 vaccination (Chi = 34.45, P = 0.000) (Table 9).

Furthermore, negative attitude toward COVID-19 vaccination was a major determinant in vaccine acceptance, with only 34.08% of pregnant women with negative attitude, compared to 53.57% and 53.85% of those with neutral and positive attitude, respectively, accepting COVID-19 immunization. (Chi = 6.74, P = 0.034) (Table 10)

## Discussions

The findings from this study on the level of Knowledge, Attitude and Acceptance towards the COVID-19 vaccine and relevant factors influencing COVID-19 vaccine acceptance among pregnant women in Tanzania's Mbeya Region highlight several key insights and challenges in vaccine uptake. The majority of responders had heard about the COVID-19

# PLOS Global Public Health

**Table 6. Univariate analysis showing the acceptance of COVID-19 vaccine.**

| Variable | COVID-19 Vaccine | | Crude OR | 95% CI | P-value |
|---|---|---|---|---|---|
| | Acceptance N (%) | Hesitancy N (%) | | | |
| **District** | | | | | |
| Mbeya urban | 50(38.78) | 98(66.22) | 0.61 | 0.521-0.812 | 0.081 |
| Chunya | 19(44.19) | 24(55.81) | 0.52 | 0.103-0.412 | 0.453 |
| Mbarali | 9(40.91) | 13(59.09) | 0.41 | 0.112-0.311 | 0.124 |
| Kyela | 6(42.86) | 8(57.14) | 0.31 | 0.423-0.851 | 0.289 |
| Rungwe | 4(66.67) | 2(33.33) | 0.16 | 0.501-0.993 | 0.078 |
| **Age group** | | | | | |
| 18-23 | 11(39.29) | 17(60.71) | 0.93 | 0.705-1.220 | 0.592 |
| 24-29 | 32(39.51) | 49(60.49) | 1.01 | 0.419-2.432 | 0.984 |
| 30-35 | 55(41.18) | 50(58.82) | 1.08 | 0.452-2.589 | 0.860 |
| 36-41 | 9(28.12) | 23(71.88) | 0.60 | 0.205-1.783 | 0.362 |
| 42-47 | 3(42.86) | 4(57.14) | 1.16 | 0.216-6.207 | 0.863 |
| **Marital status** | | | | | |
| Married | 60(40.82) | 87(59.18) | 0.81 | 0.527-1.238 | 0.327 |
| Others# | 30(34.88) | 56(65.12) | 1 | | |
| **Education** | | | | | |
| Non-formal | 7(31.82) | 15(68.18) | 0.71 | 0.582-0.875 | 0.001* |
| Primary | 9(30.00) | 21(70.00) | 0.33 | 0.134-0.826 | 0.018* |
| Secondary | 76(69.09) | 34(30.91) | 0.35 | 0.187-0.644 | 0.001* |
| College | 40(56.34) | 31(43.66) | 0.36 | 0.131-0.995 | 0.049* |
| **Occupation** | | | | | |
| Employed | 25(37.88) | 41(62.12) | 0.43 | 0.112-0.512 | 0.212 |
| Peasants | 22(38.60) | 35(61.40) | 0.21 | 0.102-0.440 | 0.101 |
| Self employed | 17(38.64) | 27(61.36) | 0.64 | 0.205-0.981 | 0.221 |
| Unemployed | 25(37.88) | 41(62.12) | 0.72 | 0.715-1.332 | 0.442 |
| **Religion** | | | | | |
| Christian | 56(45.16) | 68(54.84) | 0.53 | 0.317-0.887 | 0.016* |
| Muslim | 34(32.38) | 71(67.62) | 0.58 | 0.339-0.998 | 0.049* |
| None | 0(0.00) | 4(100.00) | 1 | | |
| **People in household** | | | | | |
| < 5 | 63(45.32) | 76(54.68) | 0.49 | 0.278-0.849 | 0.011* |
| ≥ 5 | 27(28.72) | 67(71.28) | 0.49 | 0.278-0.849 | 0.011* |
| **Children in household** | | | | | |
| < 5 | 88(38.94) | 138(61.06) | 0.63 | 0.119-3.304 | 0.582 |
| ≥ 5 | 2(28.72) | 5(71.43) | 0.63 | 0.119-3.304 | 0.582 |
| **Gestation age(weeks)** | | | | | |
| <13 | 1(14.29) | 6(85.71) | 0.75 | 0.464-1.203 | 0.230 |
| 14-27 | 40(48.78) | 42(51.22) | 5.71 | 0.658-49.590 | 0.114 |
| ≥ 28 | 49(34.03) | 95(65.97) | 3.09 | 0.362-26.432 | 0.302 |
| **Gravidity** | | | | | |
| < 5 | 84(39.88) | 129(60.56) | 0.66 | 0.243-1.780 | 0.410 |
| ≥ 5 | 6(30.00) | 14(70.00) | 0.66 | 0.243-1.780 | 0.410 |
| **Parity** | | | | | |
| Nulliparous | 29(43.28) | 38(56.72) | 0.77 | 0.442-1.338 | 0.353 |

*(Continued)*

PLOS Global Public Health

**Table 6.** (Continued)

| Variable | COVID-19 Vaccine | | Crude OR | 95% CI | P-value |
|---|---|---|---|---|---|
| | Acceptance N (%) | Hesitancy N (%) | | | |
| <5 | 60(36.81) | 103(63.19) | 0.76 | 0.428-1.362 | 0.360 |
| ≥ 5 | 1(33.33) | 2(66.67) | 0.66 | 0.057-7.582 | 0.735 |
| **Chronic medical illness** | | | | | |
| Yes | 19(63.33) | 11(36.67) | 3.21 | 1.448-7.123 | 0.004* |
| No | 71(34.98) | 132(65.02) | 3.21 | 1.448-7.123 | 0.004* |
| **High risk pregnancy** | | | | | |
| Yes | 21(35.59) | 38(64.41) | 0.84 | 0.455-1.553 | 0.020* |
| No | 69(39.66) | 105(60.34) | 0.84 | 0.455-1.553 | 0.020* |
| **Source of information** | | | | | |
| Friends/neighbors | 11(47.83) | 12(52.17) | 0.80 | 0.705-0.911 | 0.001* |
| Government leaders | 28(66.67) | 14(33.33) | 2.18 | 0.771-6.171 | 0.141 |
| Health care workers | 5(35.71) | 9(64.29) | 0.61 | 0.154-2.374 | 0.472 |
| Mass media | 13(48.15) | 14(51.85) | 1.01 | 0.333-3.084 | 0.982 |
| Social media | 24(32.88) | 49(67.12) | 2.53 | 0.068-0.633 | 0.006* |
| Religious/local leaders | 1(25.00) | 3(75.00) | 0.36 | 0.033-4.035 | 0.410 |
| Never heard | 8(19.05) | 42(80.95) | 0.21 | 0.206-1.385 | 0.197 |
| **Vaccine Knowledge** | | | | | |
| Good | 15(57.69) | 11(42.31) | 0.58 | 0.396-0.859 | 0.006* |
| Moderate | 20(48.78) | 21(51.22) | 0.70 | 0.259-1.880 | 0.477 |
| Poor | 55(33.13) | 111(66.87) | 0.36 | 0.156-0.844 | 0.019* |
| **Vaccine Attitude** | | | | | |
| Positive | 14(53.85) | 12(46.15) | 2.26 | 0.983-5.179 | 0.055 |
| Neutral | 15(53.71) | 13(46.29) | 2.23 | 0.998-4.990 | 0.050 |
| Negative | 61(34.08) | 118(65.92) | 1.61 | 1.089-2.379 | 0.017* |
| **Pharmaceuticals trust** | | | | | |
| Yes | 75(60.48) | 49(39.52) | 9.59 | 4.992-18.431 | 0.000* |
| No | 15(13.76) | 94(86.24) | 9.59 | 4.992-18.431 | 0.000* |
| **Prefer natural immunity** | | | | | |
| Yes | 66(44.30) | 83(55.70) | 0.50 | 0.284-0.892 | 0,019* |
| No | 24(28.57) | 60(71.43) | 0.50 | 0.284-0.892 | 0.019* |
| **Harm to mother's body** | | | | | |
| Yes | 28(36.36) | 49(63.64) | 0.87 | 0.493-1.523 | 0.618 |
| No | 62(39.74) | 94(60.26) | 0.87 | 0.493-1.523 | 0.618 |
| **Harm to baby** | | | | | |
| Yes | 37(38.54) | 59(61.46) | 0.99 | 0.582-1.699 | 0.982 |
| No | 53(38.69) | 84(61.31) | 0.99 | 0.582-1.699 | 0.982 |
| **Vaccinated in last 5 years** | | | | | |
| Yes | 75(83.33) | 15(16.67) | 34.72 | 16.523-72.965 | 0.000* |
| No | 18(12.59) | 125(87.41) | 34.72 | 16.523-72.965 | 0.000* |
| **Vaccinated against COVID-19** | | | | | |
| Yes | 62(96.88) | 2(3.12) | 156.11 | 36.062-675.768 | 0.000* |
| No | 28(16.57) | 141(83.43) | 156.11 | 36.062-675.768 | 0.000* |

#(Single, divorced, widow/widower)

*(P-value<0.05)

**Table 7. Multivariate analysis showing acceptance of COVID-19 vaccine.**

| Variable | COVID-19 Vaccine | | Adjusted OR | 95% CI | P-value |
|---|---|---|---|---|---|
| | Acceptance N (%) | Hesitancy N (%) | | | |
| **District** | | | | | |
| Mbeya urban | 50(38.78) | 98(66.22) | 0.84 | 0.216-6.207 | 0.120 |
| Chunya | 19(44.19) | 24(55.81) | | | |
| Mbalali | 9(40.91) | 13(59.09) | | | |
| Kyela | 6(42.86) | 8(57.14) | | | |
| Rungwe | 4(66.67) | 2(33.33) | | | |
| **Age group** | | | | | |
| 18-23 | 11(39.29) | 17(60.71) | 0.75 | 0.119-0.642 | 0.594 |
| 24-29 | 32(39.51) | 49(60.49) | | | |
| 30-35 | 55(41.18) | 50(58.82) | | | |
| 36-41 | 9(28.12) | 23(71.88) | | | |
| 42-47 | 3(42.86) | 4(57.14) | | | |
| **Marital status** | | | | | |
| Married | 60(40.82) | 87(59.18) | 0.46 | 0.215-0.711 | 0.111 |
| Others# | 30(34.88) | 56(65.12) | | | |
| **Education** | | | | | |
| Non-formal | 7(31.82) | 15(68.18) | 2.99 | 2.780-3.206 | 0.001* |
| Primary | 9(30.00) | 21(70.00) | | | |
| Secondary | 34(30.91) | 76(69.09) | | | |
| College | 40(56.34) | 31(43.66) | | | |
| **Occupation** | | | | | |
| Employed | 25(37.88) | 41(62.12) | 10.33 | 9.403-11.254 | 0.154 |
| Peasants | 22(38.60) | 35(61.40) | | | |
| Self employed | 17(38.64) | 27(61.36) | | | |
| Unemployed | 25(37.88) | 41(62.12) | | | |
| **Religion** | | | | | |
| Christian | 56(45.16) | 68(54.84) | 1.55 | 1.465-1.640 | 0.015* |
| Muslim | 34(32.38) | 71(67.62) | | | |
| None | 0(0.00) | 4(100.00) | | | |
| **People in household** | | | | | |
| < 5 | 63(45.32) | 76(54.68) | 1.47 | 1.388-1.549 | 0.011* |
| ≥ 5 | 27(28.72) | 67(71.28) | | | |
| **Children in household** | | | | | |
| < 5 | 88(38.94) | 138(61.06) | 1.03 | 1.007-1.063 | 0.581 |
| ≥ 5 | 2(28.72) | 5(71.43) | | | |
| **Gestation age(weeks)** | | | | | |
| <13 | 1(14.29) | 6(85.71) | 2.62 | 2.531-2.713 | 0.231 |
| 14-27 | 40(48.78) | 42(51.22) | | | |
| ≥ 28 | 49(34.03) | 95(65.97) | | | |
| **Gravidity** | | | | | |
| < 5 | 84(39.88) | 129(60.56) | 1.10 | 1.052-1.144 | 0.409 |
| ≥ 5 | 6(30.00) | 14(70.00) | | | |
| **Parity** | | | | | |
| Nulliparous | 29(43.28) | 38(56.72) | 1.74 | 1.670-1.827 | 0.354 |
| < 5 | 60(36.81) | 103(63.19) | | | |

*(Continued)*

**Table 7.** (Continued)

| Variable | COVID-19 Vaccine | | Adjusted OR | 95% CI | P-value |
|---|---|---|---|---|---|
| | Acceptance N (%) | Hesitancy N (%) | | | |
| ≥ 5 | 1(33.33) | 2(66.67) | | | |
| **Chronic medical illness** | | | | | |
| Yes | 19(63.33) | 11(36.67) | 1.08 | 1.023-1.131 | 0.003* |
| No | 71(34.98) | 132(65.02) | | | |
| **High risk pregnancy** | | | | | |
| Yes | 21(35.59) | 38(64.41) | 1.26 | 1.194-1.338 | 0.582 |
| No | 69(39.66) | 105(60.34) | | | |
| **Source of information** | | | | | |
| Friends/neighbors | 11(47.83) | 12(52.17) | 4.85 | 4.509-5.197 | 0.001* |
| Government leaders | 28(66.67) | 14(33.33) | | | |
| Health care workers | 5(35.71) | 9(64.29) | | | |
| Mass media | 13(48.15) | 14(51.85) | | | |
| Social media | 24(32.88) | 49(67.12) | | | |
| Religious/local leaders | 1(25.00) | 3(75.00) | | | |
| Never heard | 8(19.05) | 42(80.95) | | | |
| **Vaccine Knowledge** | | | | | |
| Good | 15(57.69) | 11(42.31) | 2.70 | 2.589-2.810 | 0.005* |
| Moderate | 20(48.78) | 21(51.22) | | | |
| Poor | 55(33.13) | 111(66.87) | | | |
| **Vaccine Attitude** | | | | | |
| Positive | 14(53.85) | 12(46.15) | 1.26 | 1.149-1.368 | 0.015* |
| Neutral | 15(53.71) | 13(46.29) | | | |
| Negative | 61(34.08) | 118(65.92) | | | |
| **Pharmaceuticals trust** | | | | | |
| Yes | 75(60.48) | 49(39.52) | 0.34 | 0.270-0.415 | 0.000* |
| No | 15(13.76) | 94(86.24) | | | |
| **Prefer natural immunity** | | | | | |
| Yes | 66(44.30) | 83(55.70) | 0.42 | 0.341-0.498 | 0.018* |
| No | 24(28.57) | 60(71.43) | | | |
| **Harm to mother's body** | | | | | |
| Yes | 28(36.36) | 49(63.64) | 0.34 | 0.265-0.420 | 0.620 |
| No | 62(39.74) | 94(60.26) | | | |
| **Harm to baby** | | | | | |
| Yes | 37(38.54) | 59(61.46) | 0.41 | 0.331-0.494 | 0.982 |
| No | 53(38.69) | 84(61.31) | | | |
| **Vaccinated in last 5 years** | | | | | |
| Yes | 75(83.33) | 15(16.67) | 0.13 | 0.068-0.183 | 0.000* |
| No | 18(12.59) | 125(87.41) | | | |
| **Vaccinated against COVID-19** | | | | | |
| Yes | 62(96.88) | 2(3.12) | 0.67 | 0.594-0.755 | 0.000* |
| No | 28(16.57) | 141(83.43) | | | |

#(Single, widow/widower, divorced)

*(P-value＜0.05)

**Table 8. Relationship between acceptance and COVID-19 vaccine knowledge level.**

| Vaccine Acceptance | Level of COVID-19 vaccine knowledge | | | Total | |
| --- | --- | --- | --- | --- | --- |
| | Good | Moderate | Poor | | |
| No | 11 | 21 | 111 | 143 | |
| | 42.31 | 51.22 | 66.87 | 61.37 | Key |
| Yes | 15 | 20 | 55 | 90 | Frequency |
| | 57.69 | 48.78 | 33.13 | 38.63 | Column |
| Total | 26 | 41 | 166 | 233 | Percentage |
| | 100.00 | 100.00 | 1000.00 | 100.00 | |

Pearson Chi²=7.88 P=0.002

**Table 9. Association between Attitude and COVID-19 vaccine knowledge.**

| Attitude to vaccine | Level of COVID-19 vaccine knowledge | | | Total | |
| --- | --- | --- | --- | --- | --- |
| | Good | Moderate | Poor | | |
| Negative | 10 | 30 | 139 | 179 | |
| | 38.46 | 73.17 | 83.73 | 76.82 | Key |
| Neutral | 5 | 6 | 17 | 28 | Frequency |
| | 19.32 | 14.63 | 10.24 | 12.02 | Column Percentage |
| Positive | 11 | 5 | 10 | 26 | |
| | 42.31 | 12.20 | 6.02 | 11.16 | |
| Total | 26 | 41 | 166 | 233 | |
| | 100.00 | 100.00 | 100.00 | 100.00 | |

Pearson Chi² = 34.45 P = 0.000

**Table 10. Relationship between Acceptance and Attitude towards COVID-19 vaccine.**

| Vaccine Acceptance | Attitude to COVID-19 vaccine | | | Total | |
| --- | --- | --- | --- | --- | --- |
| | Negative | Neutral | Positive | | |
| No | 118 | 13 | 12 | 143 | |
| | 65.92 | 46.43 | 46.15 | 61.37 | Key |
| Yes | 61 | 15 | 14 | 90 | Frequency |
| | 34.08 | 53.57 | 53.85 | 38.63 | Column |
| Total | 179 | 28 | 26 | 233 | Percentage |
| | 100.00 | 100.00 | 100.00 | 100.00 | |

Pearson Chi² = 6.74 P = 0.034

vaccination. The results showed that social media (31.33%) was the most common source of information on the COVID-19 vaccination, but reliance on it may contribute to misinformation. During the pandemic, users sought the most up-to-date information; however, effective communication is an unavoidable component of COVID-19 crisis response, and attempts to reach the public can take many forms from numerous sources.

Like other studies conducted in Tanzania, this study underscored social media platforms as the most active means to communicate health-related information about the COVID-19 vaccine due to their broad reach and extensive functionality [29]. Communication gaps exist in remote areas with limited internet access, necessitating alternative outreach strategies. Reliable and timely communication is key to the success of public health efforts [30,31]. Therefore, overcoming widespread COVID-19 vaccination hesitation necessitates a concerted public health communication strategy on which people can trust, rely, and act on [32].

Approximately 71.24% of participants had a poor knowledge of the COVID-19 vaccination. In contrast to a study conducted in Kilimanjaro, Tanzania [22] which demonstrated moderate knowledge of the COVID-19 vaccination. Knowledge has a significant impact on vaccine acceptance as those with moderate to good knowledge tend to have better understanding of the medical information, which can facilitate more informed decisions regarding vaccination [33].

It's noteworthy that 76.82% of participants expressed a negative attitude towards the COVID-19 vaccine, in line with a study conducted in Tanzania [22], where (65.52%) of participants reported negative attitude towards COVID-19 vaccine. These findings are contrast to a study done in China [34] were 91.3% had positive attitude towards COVID-19 vaccine. Negative attitudes, often fueled by misinformation or fear influenced by community norms, peer discussions and media portrayals [35], hinder vaccine uptake. It is pivot to enhance accurate and culturally sensitive health communication to leverage vaccine acceptance.

Only 38.63% of participants reported accepting the COVID-19 vaccine, which is higher than the study done among pregnant women(18.5%) attending ANC at Debre Markos, Ethiopia [25], but lower than another study done in Ethiopia 62.6% [36] Kuwait(53.1%) [37], Saudi Arabia (90.4%) [38], Middle eastern population (63.2%) [28],Bangladesh (67%) [39] and Iraq (61.7%) [40]. The discrepancies may be attributed to variations in demographic factors, health literacy, health systems and socio-economic conditions. Several factors were associated with the acceptance of COVID-19 vaccine as described below.

Higher education level was consistently associated with increased vaccine uptake (OR=0.35, CI 0.187-0.644, P=0.001). Educated individuals are more likely to access and interpret health information accurately, leading to informed decisions about vaccination [41]. Studies from both low and high income countries have shown that women with secondary or tertiary education significantly accepted COVID-19 vaccine than those with no formal education [42]. Religious teachings have led to vaccine hesitancy (OR=0.53, CI 0.317-0.887, P=0.016).Concerns about vaccine containing "haram" ingredients have been raised in predominantly Muslim communities contributing to lower vaccine uptake [43]. Chronic medical illness have heightened concern about COVID-19 complications and are often more motivated to seek for preventive measures including vaccination(AOR=3.21, CI 1.448-7.123, P=0.004). Nevertheless, paradoxically, some may fear that vaccination could worsen their conditions [44]. Understanding specific concerns of this group is essential for targeted intervention.

Women with high risk pregnancy had portrayed vaccine hesitancy (OR=0.84, CI 0.455-1.553, P=0.020).These women may be more cautious about any intervention, due to concerns over fetal outcomes [42]. Misinformation, particularly from social media can erode trust and increase hesitancy (OR=2.53, CI 0.068-0.633, P=0.006). Individuals who relied more on healthcare providers as their source of information were more likely to get vaccinated [36]. Trust in pharmaceutical companies influenced vaccine hesitancy (OR=9.59, CI 4.992-18.431, P=0.000). Skepticism towards pharmaceutical companies can hinder vaccine acceptance especially if individuals believe that companies prioritize profit over public health [45].

Preference of natural immunity over COVID-19 vaccine was associated with lower vaccine uptake (AOR=0.42. CI 0.341-0.498, P=0.018). Prior research indicated belief in natural immunity can contribute to vaccine hesitancy [46]. Education on the robust protection offered by vaccines compared to natural infection is vital in these cases. Prior vaccination history positively influences attitudes towards new vaccines (AOR=0.13, CI 0.068-0.183, P=0.000). Individuals who have regularly received vaccinations in the past are more likely to accept new vaccines like COVID-19 vaccine [47]. Additionally having being vaccinated against COVID-19 vaccine boosted more confidence to receive the vaccine (AOR=0.67, CI 0.594-0.755, P=0.000). Prior positive experience with vaccination build confidence and trust in the health system [48].

**Strength and limitations of the study**

To the best of the author's knowledge, this was the first study to report on COVID-19 vaccine Knowledge, Attitude and Acceptance levels and explore factors associated with COVID-19 vaccine among pregnant women in Mbeya. The study had several limitations that should be acknowledged to contextualize its findings and implications; First, cross sectional design which collected data at one specific moment; while it provides a snapshot of vaccine acceptability, it could not assess changes over time or establish cause and effect relationships. Second, selection bias as the study only included pregnant women who had access to ANC during the study period and agreed to participate, which may introduce selection bias. Women without ANC access (e.g., due to socioeconomic barriers, remote areas, distrust in healthcare) may have different vaccine pattern leading to underrepresentation of certain groups. Lastly, temporal validity as vaccine acceptability is influenced by evolving factors such as public trust, misinformation, policy changes and disease prevalence. This study was conducted at one point may not capture shifts due to new vaccine recommendations, emerging variants, or media influence, limiting the generalizability of findings beyond the study period.

**Conclusions**

The COVID-19 pandemic continues to devastate on lives and lively-hoods around the world however COVID-19 vaccine offers a ray of hope for the future. In Mbeya region, Tanzania there is poor knowledge of COVID-19 vaccine and negative attitudes toward COVID-19 vaccine which have resulted in the low vaccine acceptance rates. Disinformation of COVID-19 vaccine appears to be a factor related with vaccine hesitancy.

These findings emphasize that authorities should implement major educational programs, awareness campaigns and disseminate reliable and credible information about COVID-19 vaccine using media, health policymakers, researchers and stakeholders. All concerned bodies should be actively involved to help achieve higher acceptance rates of COVID-19 vaccine among pregnant women. To improve vaccine coverage, it is essential that pregnant women have sufficient knowledge about effectiveness and safety of COVID-19 vaccine.

**Supporting information**

**S1 Data.  Data checklist.**
(XLSX)

**S1 Text.  Realibity of questionnaire.**
(DOCX)

**S2 Text.  Knowledge, attitude, acceptance and associated factors.**
(DOCX)

**Author contributions**

**Conceptualization:** Revocatus Lawrence Kabanga.

**Data curation:** Revocatus Lawrence Kabanga, Vincent John Chambo.

**Formal analysis:** Revocatus Lawrence Kabanga.

**Project administration:** Vincent John Chambo.

**Supervision:** Rebecca Mokeha.

**Validation:** Vincent John Chambo, Rebecca Mokeha.

**Writing – original draft:** Revocatus Lawrence Kabanga.

**Writing – review & editing:** Vincent John Chambo.

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
