## [Decision Letter · Decision Letter 0]

PGPH-D-25-00440

Knowledge, Attitudes and Acceptance of COVID-19 Vaccine among Pregnant Women in Mbeya Region

Dear Dr. Kabanga,

Thank you for submitting your manuscript to PLOS Global Public Health. After careful consideration, we feel that it has merit but does not fully meet PLOS Global Public Health’s publication criteria as it currently stands. Therefore, we invite you to submit a revised version of the manuscript that addresses the points raised during the review process.

The reviewers have some overlapping comments. If any seem in conflict to you, or if you do not want to respond to a particular comment, just mention that in your response.

We look forward to receiving your revised manuscript.

Kind regards,

Abram L. Wagner, PhD, MPH

Academic Editor

Journal Requirements:

Additional Editor Comments (if provided):

Reviewers' comments:

Reviewer's Responses to Questions

**Comments to the Author**

1. Does this manuscript meet PLOS Global Public Health’s publication criteria ? Is the manuscript technically sound, and do the data support the conclusions? The manuscript must describe methodologically and ethically rigorous research with conclusions that are appropriately drawn based on the data presented.

Reviewer #1: Partly

Reviewer #2: Yes

Reviewer #3: Partly

Reviewer #4: Yes

Reviewer #5: No

Reviewer #6: Yes

Reviewer #7: No

2. Has the statistical analysis been performed appropriately and rigorously?

Reviewer #1: I don't know

Reviewer #2: Yes

Reviewer #3: Yes

Reviewer #4: Yes

Reviewer #5: I don't know

Reviewer #6: No

Reviewer #7: Yes

3. Have the authors made all data underlying the findings in their manuscript fully available (please refer to the Data Availability Statement at the start of the manuscript PDF file)?

Reviewer #1: No

Reviewer #2: Yes

Reviewer #3: No

Reviewer #4: No

Reviewer #5: Yes

Reviewer #6: Yes

Reviewer #7: No

4. Is the manuscript presented in an intelligible fashion and written in standard English?

Reviewer #1: Yes

Reviewer #2: Yes

Reviewer #3: No

Reviewer #4: Yes

Reviewer #5: No

Reviewer #6: No

Reviewer #7: Yes

5. Review Comments to the Author

Reviewer #1: In this cross-sectional analysis, the authors attempt to assess COVID-19 vaccine knowledge, attitude, and acceptance among pregnant women in one region of Tanzania. They find low vaccine knowledge and acceptance, with primarily negative attitudes towards the vaccine. While I do think there is potential merit here, especially in understanding vaccine hesitancy in a critical population, this manuscript would require substantial revision before it is acceptable for publication. The majority of my concerns are in the Methods.

Introduction:

1. The authors say in the Introduction (for example: see Broad Objective) that their goal with this work is to increase awareness of the COVID-19 vaccine among pregnant people. However, the study doesn't appear to be doing that so much as measuring current levels of awareness. If the authors did do any kind of health education or patient outreach on the vaccine, this should be made clear.

2. I suggest working the last two paragraphs of the Problem Statement directly into the Introduction, as well as the Objectives, Hypothesis, Research Questions, and Significance of the Study sections. In doing so, it would also be useful to focus in on the specific context that the authors are studying (Mbeya) and present a clearer picture of their overall aim. By having each of these sections separate, rather than integrated into the Introduction, it is difficult to follow a cohesive through-line.

3. Please provide a citation for the line "Pregnant women may be prone to COVID-19 vaccine..." in the last paragraph before the Problem Statement header

Methods

1. Why Mbeya region? The reason for the specific focus on this region should be made clear. Likewise, why was MZRH chosen?

2. In the "Exclusion criteria" section, how was “with mental illness” defined? Was this self-reported? Why were they excluded?

3. It is not clear how or when the participants were selected, randomized, or given the survey. Did the authors pull from a list of patients from the facility who had appointments during the study period? Were they approached at their appointment? How was the survey administered (online, in person, etc.)? This part of the study requires much more detail. It would also be helpful if the authors could include the survey in the supplemental materials.

4. How was the questionnaire created? Was it based on a standard survey for vaccine hesitancy? Was it validated through any means? Says “questionnaire was well prepared” and “pre-tested,” but can we have more details there?

5. Did all of the participants speak Swahili? Was it a requirement?

6. How were incomplete surveys and data missingness handled?

7. There also needs to be more information about the logistic regression the authors discuss using. Were these univariate or multivariate? What variables were included? Are the dependent and independent variables listed in the Methods the ones used in the models? The results of the models (i.e., the odds ratios), should be presented in the main text.

8. What do the authors mean by "correct" and "incorrect" response for the attitude questions? Some of the questions, like whether they feel as though the government is making decisions in their best interest, don't necessarily have a right and wrong answer.

9. Please define MZRH at first use

10. The exact sample size calculation can either be removed, shortened, or moved to the appendix.

Results

1. As I mention previously, the logistic regression results should be presented here. Tables 6 - 8 just present frequencies and percentages rather than the ORs and 95% CIs that we would expect from a logistic regression.

2. For the logistic regression results, why did the authors use the “If the COVID-19 vaccine are recommended for pregnant women, will you get vaccinated” question to determine acceptance rather than the number of women who had actually already gotten the vaccine?

3. The authors briefly mention that having a trusted source of information was a determinant of COVID-19 vaccine acceptability. What question was used to determine whether they did? Same with regard to "attitude towards COVID-19 vaccine"

Discussion

1. There should be a bit more nuance with how social media is treated as an information source. It is often referred to as a "trusted" or "reliable" source, but there should be appropriate caveats throughout that the information gathered via social media may not be correct as this was not explicitly measured as part of the study.

2. In the Discussion, the authors should spend more time interrogating the study results rather than simply restating them. It isn't very useful to compare these findings to broad findings from other countries, as the contexts are likely not comparable. Instead, can the authors situate their results more in the Tanzania context and discuss interesting (and unexpected) findings? Also, the limitations section should be moved from the Methods section to the Discussion and should be heavily expanded.

Reviewer #2: It appears a good study The objective of the study should be specific and the title should reflect the objective. A lot of unwanted things are dragged and discussed which can be excluded from the study.

Reviewer #3: The authors presented good findings on COVID-19 vaccine acceptance and hesitancy among pregnant women, which have a public health impact.

However, the authors may improve their manuscript by

1) Avoiding the use of non-standard abbreviations such as MZRH, ANC, RCH, and UDSM-MCHAS throughout the article without first defining them.

2) Some sentences are not complete and lack clarity. Please revise the last sentence of the result section in the abstract.

3) To increase the reliability of the finding, more description that is detailed in the Material and Method section is needed.

Other Comments

1) Please insert a reference for this sentence: “Pregnant women may be prone to COVID-19 vaccine disinformation particularly anti-vaccination buzz as well as lack of reliable information due to social marginalization and language barriers…” in the background section.

2) Considering the data analysis conducted, this study looks descriptive with an analytical component. Please revise the design section.

3) The Duration of the study period reported appears to be only five days. Does the reported period entail study duration or data collection period?

4) The study area is well presented but it is not justified for the objective of this manuscript.

5) Minimum calculated sample size was 233 and a total of 233 pregnant women were interviewed, but in ensuring study validity, the authors stated that maximum sample sizes were employed. For clarity and accuracy, it would be helpful to rephrase this sentence.

6) The questionnaire was assessed by the supervisor but there is no description of who the supervisor was and their qualifications. To ensure transparency, it would be helpful to describe the supervisor's role in this manuscript.

7) In the result section, some percentages in most tables do not add up to 100%. Please revise Tables 1 and 2

8) The final paragraph of the discussion section seems to be a result rather than a critical discussion of these results.

Reviewer #4: Very fascinating study. I have a couple of concerns:

1. Did participants only indicate one source of information? I find that to be unlikely. Also, your Figure 2 show the percentage of never heard to be 21.46% while Figure 1 shows the percentage of 'no' to be 24%. Is there any reason for this difference?

2. Under your ‘source of information on COVID-19 vaccine’ section, you interpreted your results as - 'social media was the most trusted source of information on COVID-19 vaccine, accounting for 31.33%'. However there is a difference between something being a SOURCE and it being a TRUSTED SOURCE. Which of this exactly did your study measure? Did people say social media was their SOURCE of information or did they say it was their TRUSTED SOURCE of information? If it is the later, then you have to update the labels in your charts to reflect that.

3. Please rewrite the first paragraph under the Acceptance to COVID-19 vaccine section for clarity.

4. Please format your manuscript for punctuation and capitalization of sentences.

5. The last paragraph in your discussion section reads 'Education level, religion, chronic medical illness, high risk pregnancy, having heard about COVID-19 vaccine, trusted source of information, trusting pharmaceutical companies, preference to natural immunity than COVID-19 vaccine, harm to the baby after delivery, having vaccinated for the past five years, attitude towards COVID-19 vaccine and the knowledge of COVID-19 vaccine were the determinants of vaccine acceptance.' However, you did not perform analysis for all of the factors you mentioned here so you cannot categorically say they all are determinants unless you add proper citations. Please consider this.

6. In the limitations section, please consider acknowledging that your study does not address several material and contextual factors that may influence vaccine acceptance. These include:

Logistical/material barriers such as transportation, availability and access to vaccines;

Time-related constraints, such as the ability to take time off work;

Social and cultural influences, including prevailing social norms and whether individuals observe vaccine uptake among peers or within their networks.

Acknowledging these factors would provide a more comprehensive view of the limitations of your study and clarify the scope of your findings.

Reviewer #5: Thank you for giving me an opportunity to review this study.

General

- The paper will require a general review of placement of full stops and some grammar. Not consistent how references are placed. Ideally, it should be after a full stop with no full stop after it.

- The paper reads more like a protocol than a research paper stylistically, especially the first half. The authors should consider, for example, consolidating most of the subheadings under Materials and Methods into paragraphs instead of distinct, different headings.

Key Terms

- Authors to consider revising the definition of COVID 19 to pathogenic viral infection caused by severe acute respiratory syndrome coronavirus 2 (SARS-CoV-2).

Introduction

- The authors may consider adding, apart from the arguments regarding safety of COVID-19 vaccines in pregnant women, that the vaccines have been recommended in women planning to get pregnant, pregnant and breastfeeding women and the WHO has authorized the vaccines in pregnant women.

- “Pregnant women may be prone to COVID-19 vaccine disinformation particularly anti-vaccination buzz as well as lack of reliable information due to social marginalization and language barriers”. Pregnant women would not be the only demographic that may be susceptible to vaccine disinformation.

- Authors may want to consider integrating the information in the Problem statement into the Background section as it seems the broad concepts in both sections are similar.

- Because there seemed to be more concern over the safety and efficacy of some COVID vaccines over some others, the authors may consider adding to this section which COVID vaccines were available in Tanzania as at the time of conducting the study.

Hypothesis

- For the 2nd hypothesis; “Disinformation regarding COVID-19 vaccine among pregnant women contribute to its lower acceptance rate in Mbeya region.” It may be that the study will show that disinformation will contribute to lower acceptance among pregnant women and not the whole Mbeya population?

Exclusion criteria

- Authors to consider reformulating the exclusion criteria to make it clearer. In its current state, it reads as though pregnant women that were too ill, unable to hear or read AND had mental illness were excluded.

- And why was lack of hearing in particular an exclusion criteria? And was it a lack of hearing and reading in a particular language? And as they had to sign a written informed consent form, what happened in cases where they could not write?

Dependent variables

- I am not familiar with Bloom’s scoring. However, should the Bloom’s cut off point for a score <60% not be <5.4 as against 5.4 since that of 60-79% is 7.1-5.4? And also please explain what the values you have put in the brackets mean.

- In structuring the headings in this section, it would make it easier to understand if the variable itself (as against the variable and it’s classifications) is given as a heading, then the various classifications of a variable be described in the text that follows. For example, “Having knowledge of COVOD-19” as a heading and then describing what the good, moderate or poor knowledge classifications are.

- While discussing attitude to COVID-19, what was the rational for writing the Bloom’s cut offs in descending order. For example, 100-80%, instead of 80-100%. And the same question for the figures in brackets.

Ethical issues

- Please define all the abbreviations used in this section as they appear for the first time here.

Study limitations

- Propose that this section be moved to after the discussion.

- Could social desirability have played a role in some of their responses?

- What would the authors consider as strengths of their study?

Questionnaire

- The description of the questionnaire is missing a lot of details such as how was it developed, what language was it in originally, was it translated into a local language and what steps did the investigators take to ensure that the translation was correct, were there steps to ensure content validity, how was the questionnaire tested before it was put to final use (was it for example tested on potential participants and were these then excluded form the final study pool), was the questionnaire self-administered or administered by an investigator etc?

Results

- A response rate of 100% was given. Does that mean that all the women that were approached agreed to participate in the study? And there were no women that were approached that met the exclusion criteria?

- In the table, under Occupation, please define HCW.

- Table 2: How were high risk pregnancies defined in the study?

- Table 3: For the question “Do you have enough evidence about COVID-19 vaccine safety during pregnancy?”, it’s difficult to understand how the responders would have properly responded to this question. Or was the intention to ask if they know, generally, if there is enough evidence on COVID-19 vaccine safety?

- Tables 6, 7 and 8: The keys/legends you have added to the sides of the tables are a bit confusing. Authors to consider simply putting the 2 figures on one line as “Total (%)”. For example, for relationship between vaccine non-acceptance and knowledge, present the figure as 11 (42.31%). And for the same tables, were the p-values exactly the same for all the variables?

Discussion

- This section will benefit from a more nuanced discussion of the issues. For example, when discussing COVID-19 acceptance, the authors compared their acceptance rates with others done in other countries. However, I don’t believe the conclusions given by the authors at the end of that paragraph is enough. For example, what could be the possible reason that the acceptance in the study in Ethiopia was very low (apart from difference in population and socio-economic status)? If you use population and socio-economic status as a reason, how then would you explain the comparison you made Saudi Arabia and Iraq? Saudi Arabia is comparable to Kuwait socioeconomically, but why do they have very different acceptance rates? Another example is the final paragraph in the discussion. The authors have simply re-stated the results of their study but have not necessarily discussed or contextualized these results.

Reviewer #6: The authors present an original and timely study assessing knowledge, attitudes, and acceptance of COVID-19 vaccines, as well as the associated factors. This topic is of significant public health relevance, particularly in the current climate of eroded trust in science and public health institutions. Although there are areas that require revision and clarification, the reviewer believes the manuscript holds value and merits publication following appropriate revisions.

The manuscript would benefit from substantial language revision to enhance clarity. While the reviewer typically refrains from such comments, the prevalence of ambiguous and imprecise language throughout the manuscript often obscures the authors' intended meaning.

For instance, in the abstract, the sentence "acceptance of the vaccine was low by 38.63%" is unclear. Does this indicate a 38.63% acceptance rate or a reduction of 38.63% in acceptance? Such ambiguity should be avoided.

Furthermore, when discussing associations between factors and outcomes (knowledge, attitude, and acceptance), the manuscript should clearly delineate which factors are associated with which specific outcomes. Combining these elements into a single sentence may confuse readers. For example, it is unclear whether perceived risks to infants post-delivery are associated with basic knowledge, attitude, or acceptance.

The methodology section is generally well described and provides sufficient detail to support reproducibility. However, there is inconsistency in the terminology used to describe the data collection tool—at one point it is referred to as a structured questionnaire, and elsewhere as a semi-structured questionnaire (unfortunately, the lines are not numbered, to point out the exact places). This discrepancy should be corrected.

Regarding the statistical analysis, while the bivariate analysis is adequately described, the logistic regression lacks critical description. Specifically, the authors should:

1- Clearly describe how variables were selected and coded for inclusion in the model.

2- Indicate whether regression model assumptions were tested and met.

3- Explain how model fit was assessed.

4- Report the regression coefficients, odds ratios, and 95% confidence intervals for the final model in the results section.

The discussion section tends to repeat findings from the results rather than offering deeper interpretation or broader contextualization. The authors are encouraged to reduce redundancy and use the discussion to address study limitations, and relevance.

One important limitation not addressed in the manuscript is the potential for selection bias, given that the study population was drawn from a hospital setting. Hospital-based participants may be more inclined to accept vaccines or possess higher levels of health literacy than the general population. This should be explicitly acknowledged and discussed.

The reviewer encourages the authors to strengthen the manuscript by providing these details and clarification.

Reviewer #7: For the authors, I have the following comments:

This paper needs to present some context on Covid vaccination information and availability in TZ. I know from personal experience in TZ that initially the govt denied the utility of the vaccine and was delayed in placing orders etc. While that perspective did change over time, it would be very important to provide this background and context for this study.

The study as conducted during a 6 day period in one hospital. While they do provide information on how sample size was determined, there isn’t information on how that population represents the larger population of the region? Who comes to the hospital for ANC? Who doesn’t come there or doesn’t seek ANC? This information is necessary to understand any bias. The limitations mentions some of these issues but doesn’t not discuss how any bias could impact results.

While ‘social media’ is mentioned as a primary source of information, there are not details as to what this means. What type of social media? What sites or apps? What is the gov’t doing to promote (or not) the vaccine and are they using any of these mechanisms? Are any ‘official’ communications methods mentioned? There is a serious lack of detail on this key aspect of the study.

Some of the questions in the survey seemed a bit leading such as asking about whether the vaccine caused infertility or other harm. There could be bias introduced through the wording of the questions and this is not addressed in the narrative.

There were a few grammatical and punctuation errors in the paper which should be corrected.

6. PLOS authors have the option to publish the peer review history of their article (what does this mean? ). If published, this will include your full peer review and any attached files.

**Do you want your identity to be public for this peer review?** For information about this choice, including consent withdrawal, please see our Privacy Policy .

Reviewer #1: No

Reviewer #2: **Yes: ** Dr. Tarana Jahan

Reviewer #3: No

Reviewer #4: No

Reviewer #5: **Yes: ** Grace Mambula

Reviewer #6: **Yes: ** Kenny Moise

Reviewer #7: No

---

## [Editor Report · Decision Letter 1]

PGPH-D-25-00440R1

Knowledge, Attitudes and Acceptance of COVID-19 Vaccine among Pregnant Women in Mbeya Region

Dear Dr. Kabanga,

Thank you for submitting your manuscript to PLOS Global Public Health. After careful consideration, we feel that it has merit but does not fully meet PLOS Global Public Health’s publication criteria as it currently stands. Therefore, we invite you to submit a revised version of the manuscript that addresses the points raised during the review process.

We look forward to receiving your revised manuscript.

Kind regards,

Abram L. Wagner, PhD, MPH

Academic Editor

Editor comments:

Can you provide a citation for: "The COVID-19 pandemic impacted numerous countries globally, including those in Africa. Africa was the last continent affected by the pandemic and was anticipated to be the region where the disease would spread rapidly and exert significant impact." In particular - *who* anticipated it to be the region with the most impact?

can you move study limitations to the discussion prior to the conclusion?

I think you can have either the specific objectives or the research questions - both seem redundant.

I would delete Figure 1 - I don't think pie charts provide much information.

For Figure 2 - could you re-order form highest to lowest?

Results - when you mention age in the paragraph, can you put in "years" as well?

Note in terminology - "peasant" in North American English can seem to be derogatory. You are free to keep that, especially if that is the term people use to self identify, but "farmer" or the like seems potentially more interpretable (?)

In the intro, you have "A study conducted in Tanzania indicated that COVID-19 vaccine hesitancy was approximately 65%" - what did they mean by 65% vaccine hesitant? Was that how many people refused a vaccine, or were questioning it? Different studies have different definitions of vaccine hesitancy, so can be good to clarify.

I do not think Figure 3 adds much. I only include Figures if they include more information beyond what you can say in a paragraph text. I could maybe think of somehow including a bar chart of the different knowledge questions, and how often people responded correctly to them?

Figures 4-6 also don't add much to the article.

I would personally prefer you to write out KAA throughout the manuscript instead of relying on an acronym.

In the limitations in discussion, I would make that narrative instead of using bullet points.

---

## [Editor Report · Decision Letter 2]

Knowledge, Attitudes and Acceptance of COVID-19 Vaccine among Pregnant Women in Mbeya Region

PGPH-D-25-00440R2

Dear Acceptance of COVID-19 vaccine Kabanga,

We are pleased to inform you that your manuscript 'Knowledge, Attitudes and Acceptance of COVID-19 Vaccine among Pregnant Women in Mbeya Region' has been provisionally accepted for publication in PLOS Global Public Health.

Best regards,

Abram L. Wagner, PhD, MPH

Academic Editor